# An Aeromagnetic Compensation Algorithm Based on Radial Basis Function Artificial Neural Network

Shuai Zhou [1], Changcheng Yang [1], Zhenning Su [2], Ping Yu [1] and Jian Jiao [1,*]

[1] College of GeoExploration Science and Technology, Jilin University, Changchun 130012, China
[2] Institute of Geophysical and Geochemical Exploration, Chinese Academy of Geoscience, Langfang 065000, China
* Correspondence: jiaojian001@jlu.edu.cn

**Abstract:** Aeromagnetic exploration is a magnetic exploration method that detects changes of the earth's magnetic field by loading a magnetometer on an aircraft. With the miniaturization of magnetometers and the development of unmanned aerial vehicles (UAV) technology, UAV aeromagnetic surveying plays an increasingly important role in mineral exploration and other fields due to its advantages of low cost and safety. However, in the process of aeromagnetic measurement data, due to the ferromagnetic material of the aircraft itself and the change of flight direction and attitude, magnetic field interference will occur and affect the measurement of the geomagnetic field by the magnetometer. The work of aeromagnetic compensation is to compensate for this part of the magnetic interference and improve the magnetic measurement accuracy of the magnetometer. This paper focused on the problems of UAV aeromagnetic survey data processing and improved the accuracy of UAV based aeromagnetic data measurement. Based on the Tolles–Lawson model, a numerical simulation experiment of magnetic interference of UAV-based aeromagnetic data was carried out, and a radial basis function (RBF) artificial neural network (ANN) algorithm was proposed for the first time to compensate the aeromagnetic data. Compared with classical backpropagation (BP) ANN, the test results of the synthetic data and real measured magnetic data showed that the RBF-ANN has higher compensation accuracy and stronger generalization ability.

**Keywords:** aeromagnetic compensation; radial basis function; deep learning; unmanned aerial vehicles (UAV); local minimum

## 1. Introduction

With the development of the global economy, the demand for mineral resources in all countries in the world is also increasing. However, due to complex terrain conditions, many areas rich in mineral resources cannot be explored. In order to increase the detection range and improve exploration efficiency, aeromagnetic measurement technology has been rapidly developed. Airborne magnetic surveying is an important airborne geophysical exploration method, which can be used for magnetic data acquisition under various complex terrain conditions.

Moreover, UAV technology has developed very rapidly and has been well used in all walks of life, so UAV survey technology has gradually developed, and is now widely used in resource exploration, regional survey and other fields [1,2]. With the development of UAV technology, more and more countries have carried out the research and development of UAV aeromagnetic measurement equipment technology and achieved remarkable results. The available information indicates that the first company in the world to develop UAV aeromagnetic survey equipment was Magsurvey in the United Kingdom, which developed the PrionUAV aeromagnetic survey system in 2003 [3]. Since then, many companies around the world have conducted research and development of UAV aeromagnetic survey systems, such as the GeoRanger-I of the Dutch company Fugro [4], the Canadian company Universal Wing Geophysical (UWG) Venturer [5], the Japanese RMAX-G1 [6], the Swiss and German

jointly developed reconnaissance B1-100 [7], the German MD4-1000 [8], CH-3 [9] from the Institute of Geophysical and Geochemical Exploration (IGGE) of the Chinese Academy of Geological Science, and an integrated multi-rotor aeromagnetic survey system at Queen's University in Canada [10].

Due to the ferromagnetic material inside the UAV and its various influences during flight, it will inevitably cause certain interference to the data collected by the magnetometer sensor; if we want to obtain high-quality aeromagnetic data, we must study the appropriate aeromagnetic compensation technology [11,12].The compensation methods of aeromagnetic interference are mainly divided into hardware compensation and software compensation. The hardware compensation method is to first calculate the magnetic interference of the detection platform, and then add several coils to the detection platform to counteract the magnetic interference generated by the aircraft. In the late 20th century, high-cost, low-precision hardware compensation began to be slowly replaced by software compensation [11]. According to the nature and causes of magnetic interference, Tolles and Lawson divided it into constant interference, induced interference, and eddy current interference, and established the classic Tolles–Lawson model (T-L model) [13], which is the foundation on which current aeromagnetic compensation methods are built. Based on the T–L model, Leliak established an aeromagnetic compensation method based on FOM compensation flight [14]. The variables in the T–L equation are not independent of each other, and the linear relationship between the variables affects the stability of the solution, and the linear relationship between the variables is called multicollinearity. Bickel proposed a small-signal method to weaken the linear relationship between variables, resulting in a more stable solution [15]. Leach first proposed to overcome the multicollinearity problem of equations by introducing regularization terms through the linear regression method [16], Hardwick et al. proposed a compensation algorithm for total field gradients [17]. Dou proposes a new real-time method based on recursive least squares, and the simulation results showed that the method has a good ability to compensate for magnetic interference caused by an aircraft and its maneuvering [18]. Wu et al. use principal component analysis (PCA) to reduce the multicollinearity of the T-L model [19]. Xu applied deep learning to magnetic anomaly detection and noise cancellation [20].

Considering the lack of computational accuracy and generalization ability of linear regression methods, people began to explore new aeromagnetic compensation algorithms through neural networks. Williams successfully established an aeromagnetic compensation model based on a neural network for the first time, but his model had the problem of overfitting [21]. Zhang proposes a new compensation method that used a one-dimensional convolutional neural network to perform secondary compensation on the data that were compensated by the T–L model to eliminate the influence of tail boom swing, which has a significant compensation effect on aeromagnetic noise [22]. Ma proposed a dual estimation method for aeromagnetic compensation, combining a linear model with a neural network to improve the accuracy of magnetic compensation [23]. Although the above two methods improved the accuracy of aeromagnetic compensation, there were also problems, such as difficult parameters selection and complex network structures in design. Yu et al. proposed an aeromagnetic compensation algorithm based on deep autoencoder (DAE) [24], which reduced the multicollinearity between variables in the T–L equation, but it was not perfect for the feature extraction of high-dimensional complex data in the training process of the autoencoder network, and the local minimum problem easily occurred. They then proposed to use a generalized regression neural network (GRNN) to establish an aeromagnetic compensation model, which had a fast calculation speed, high compensation accuracy, and no backpropagation [25]. Although they solved the problem of over fitting, the problem of gradient disappearance was not considered.

The aeromagnetic compensation method based on a neural network still has some problems to be solved, such as local minimum problems in backpropagation, difficult parameters selection, and complex network structures. In order to further improve the accuracy of aeromagnetic compensation, this paper proposed, for the first time, a magnetic

compensation method based on BRF-ANN, which is widely used in function approximation, pattern recognition, and signal processing [26]; it is also widely used in aerospace, such as the longitudinal channel flight control of small UAVs [27], the navigation of UAVs [28], issues related to the optimization of UAV [29], and so on. The hidden nodes of RBF-ANN adopted the distance between the input mode and the center vector (such as the Euclidean distance) as the independent variable of the function and used the radial basis function (such as the Gaussian function) as the activation function, which is a local approximation network with better generalization ability and a simple network design. The paper is divided into five chapters. The first and last chapters present the introduction and conclusion. The second chapter discusses compensation models and methods, introducing the T–L model and the principles of BP-ANN and RBF-ANN, including the characteristics of RBF-ANN. The third chapter introduces data simulation and testing, and the fourth chapter shows the testing of the measured data; the results are displayed in graphs and tables and a method effectiveness analysis was also carried out. The application and analysis of theoretical synthetic data and real measured aeromagnetic compensation data showed that the proposed method effectively solved the problem of high-precision compensation of aeromagnetic survey data based on rotary wing UAV platform, and greatly improved the error compensation accuracy of aeromagnetic dynamic measurement data.

## 2. Compensation Models and Methods

### 2.1. T-L Model

The conversion relationship between the local coordinate system and the aircraft coordinate system is shown in Figure 1. The center of the magnetometer probe mounted on the drone is set as the origin of the coordinate system $O$, $x_b$, $y_b$, and $z_b$ are the coordinate axes of the aircraft coordinate system, and their distribution direction is parallel to the direction of the three axes of the magnetometer. $x$, $y$, and $z$ are the spatial axes of the local coordinate system at the same origin as the aircraft coordinate system. The $y_c$ axis is the projection of the $y_b$ axis on the $xOy$ plane. The flight attitude during the flight of the drone can be divided into three parts: side sliding, roll and pitch. Where, is the angle between xb axis and plane $xOy$, is the angle between $y_b$ axis and $y_c$ axis, and is the angle between $y$ axis and $y_c$ axis. The local coordinate system can be rotated according to the sequence of yaws $\psi$, pitches $\lambda$ and rolls $\theta$. The rotation sequence cannot be replaced at will [5].

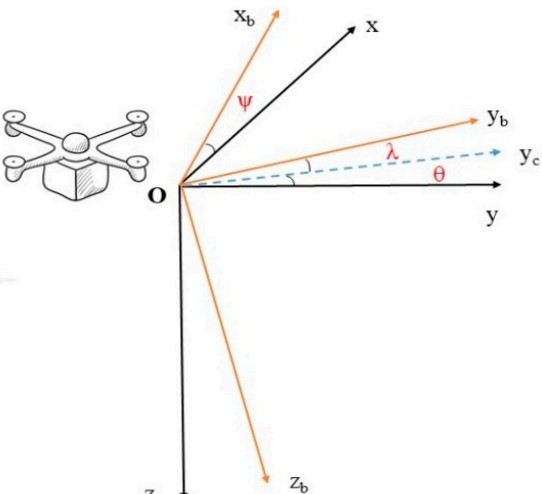

**Figure 1.** Coordinate system conversion relationship.

Tolles and Lawson divide interference fields into constant interference fields, induced interference fields, and eddy current interference fields, according to their nature and causes [9]. The constant interference field ($H_p$) is caused by direct current in the ferromagnetic material and wires inside the aircraft, and its value and direction are independent of

the flight attitude of the aircraft, and the interference value is constant for the same aircraft. The induced interference field ($H_i$) is produced by the magnetization of soft magnetic or paramagnetic substances inside the aircraft by the geomagnetic field, and its magnitude and direction are closely related to the flight attitude of the aircraft and the changes of the local geomagnetic field. The eddy current interference field is generated by the metal body cutting the geomagnetic field magnetic induction line during the flight of the aircraft, and its size and direction change with the change of the geomagnetic field gradient, flight acceleration and flight action [5]. They can be represented in the aircraft coordinate system as:

$$H_P = c_1 u_1 + c_2 u_2 + c_3 u_3 \tag{1}$$

$$H_i = T(c_4 u_1^2 + c_5 u_1 u_2 + c_6 u_1 u_3 + c_7 u_2^2 + c_8 u_2 u_3 + c_9 u_3^2) \tag{2}$$

$$\begin{aligned} H_{ec} = T(&c_{10} u_1 u_1' + c_{11} u_2 u_1' + c_{12} u_3 u_1' + c_{13} u_1 u_3' \\ &+ c_{14} u_2 u_3' + c_{15} u_3 u_3' + c_{16} u_1 u_2' + c_{17} u_2 u_2' + c_{18} u_3 u_2') \end{aligned} \tag{3}$$

$$u_1 = \frac{T_{bx}}{T_t}, u_2 = \frac{T_{by}}{T_t}, u_3 = \frac{T_{bz}}{T_t} \tag{4}$$

$$T_t = \sqrt{T_{bx}^2 + T_{by}^2 + T_{bz}^2}, \tag{5}$$

where $T$ represents the geomagnetic field, $c_1$, $c_2$, ......, $c_{18}$ represents the compensation coefficient, $u_1$, $u_2$, and $u_3$ are the cosine values of the angle between the three axes of the aircraft coordinate system and the geomagnetic field direction, $u_1'$, $u_2'$, $u_3'$ are the differentiation of $u_1$, $u_2$, $u_3$ with respect to time $t$, $T_t$ represents the total geomagnetic field data measured by the optical pump magnetometer, $T_{bx}$, $T_{by}$ and $T_{bz}$ represent the three components of the fluxgate. The accuracy of the measured data of the triaxial fluxgate magnetometer is far inferior to that of the optical pump magnetometer, and it also needs to be corrected accordingly during its installation, resulting in errors in the measured fluxgate three-component data, which indirectly affects the compensation effect. Therefore, this paper needs to make corresponding corrections to the measured fluxgate three-component data, and the correction formula is as follows:

$$\begin{bmatrix} T_{bx} \\ T_{by} \\ T_{bz} \end{bmatrix} = D \begin{bmatrix} T_{gx} \\ T_{gy} \\ T_{gz} \end{bmatrix} \tag{6}$$

$$D = \begin{bmatrix} \cos\theta\cos\psi & \sin\theta\cos\psi & -\sin\psi \\ \cos\theta\sin\lambda\sin\psi - \sin\theta\cos\lambda & \sin\theta\sin\lambda\sin\psi + \cos\theta\cos\lambda & \sin\lambda\cos\psi \\ \cos\theta\cos\lambda\sin\psi + \sin\theta\sin\lambda & \sin\theta\cos\lambda\sin\psi - \cos\theta\sin\lambda & \cos\lambda\cos\psi \end{bmatrix} \tag{7}$$

$$T_{gx} = T\cos\varphi\cos\mu \tag{8}$$

$$T_{gy} = T\cos\varphi\sin\mu \tag{9}$$

$$T_{gz} = T\cos\varphi\sin\varphi, \tag{10}$$

where $\Psi$ is the side roll angle when the aircraft is flying, $\lambda$ is the pitch angle when the aircraft is flying, $\theta$ is the side slip angle when the aircraft is flying, $\mu$ is the magnetization bias angle, and $\varphi$ is the magnetization tilt angle

Finishing Formulas (1)~(5) can be obtained, the total interference magnetic field ($H_t$) of the aircraft is:

$$\begin{aligned} H_t &= H_p + H_i + H_{ec} \\ &= T(c_1 u_1/T + c_2 u_2/T_t + c_3 u_3/T \\ &+ c_4 u_1^2 + c_5 u_1 u_2 + c_6 u_1 u_3 + c_7 u_2^2 + c_8 u_2 u_3 + c_9 u_3^2 \\ &+ c_{10} u_1 u_1' + c_{11} u_2 u_1' + c_{12} u_3 u_1' + c_{13} u_1 u_3' + c_{14} u_2 u_3' \\ &+ c_{15} u_3 u_3' + c_{16} u_1 u_2' + c_{17} u_2 u_2' + c_{18} u_3 u'), \end{aligned} \tag{11}$$

According to the relationship between $u_1$, $u_2$, and $u_3$

$$\begin{aligned} u_1^2 + u_2^2 + u_3^2 &= 1 \\ u_1 u_1' + u_2' u_2' + u_3 u_3' &= 0 \end{aligned} , \tag{12}$$

After sorting out Equations (11) and (12), we get:

$$H_t = \begin{bmatrix} b_1 \\ b_2 \\ b_3 \\ b_4 \\ b_5 \\ b_6 \\ b_7 \\ b_8 \\ b_9 \\ b_{10} \\ b_{11} \\ b_{12} \\ b_{13} \\ b_{14} \\ b_{15} \\ b_{16} \end{bmatrix}^T \begin{bmatrix} u_1/T \\ u_2/T \\ u_3/T \\ u_1^2 \\ u_1 u_2 \\ u_1 u_3 \\ u_2 u_3 \\ u_2^2 \\ u_1 u_1' \\ u_2 u_1' \\ u_3 u_1' \\ u_1 u_3' \\ u_2 u_3' \\ u_3 u_3' \\ u_1 u_2' \\ u_3 u_2' \end{bmatrix} , \tag{13}$$

where $b_1$, $b_2$, ..., $b_{16}$ represent 16 compensation coefficients.

### 2.2. BP Artificial Neural Network

BP-ANN is an algorithm that can learn and store the relationship between input data and output data without knowing the relationship between the two; it is currently the neural network with the highest application frequency and the widest application field. The calculation process of BP-ANN mainly consists of two parts: forward propagation of information and backpropagation of error. The process of forward propagation is to conduct the input data in the order of the input layer, the hidden layer, and the output layer, and then compare the output data with the expected output. If the error reaches the specified range, or the number of training times reaches a certain number of times, the training can be stopped, otherwise it will be transferred to the backpropagation process of error. The backpropagation of error refers to the process of finding the parameters corresponding to the minimum value of the loss function of the neural network by continuously and iteratively optimizing the weights and biases in the neural network. At present, gradient descent is the most widely used optimization method in the backpropagation process. The BP-ANN structure diagram is shown in Figure 2.

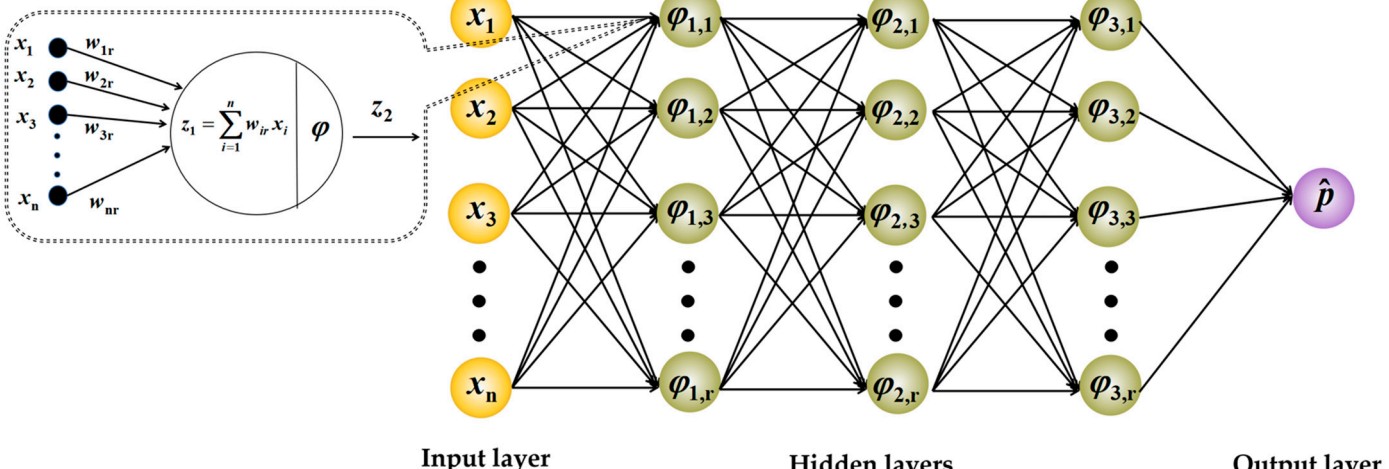

**Figure 2.** Structure diagram of BP-ANN.

In the process of aeromagnetic compensation, the causes of aeromagnetic interference were analyzed to determine the training parameters of BP-ANN, the same goes for RBF-ANN, the compensation model had 9 input parameters, which consists of the fluxgate three components, the directional cosine and its derivative to time. In addition, the output layer is magnetic interference ($H_t$), and the number of hidden layers of BP-ANN and the number of nodes per hidden layer can be determined by trial and error and the following empirical formula:

$$r = \sqrt{nl} + \delta,\tag{14}$$

where $\delta$ take the integer (experience value) of 1~10, $r$ is the number of hidden layer nodes, $n$ is the number of input layer nodes, $l$ is the number of output layer nodes. In this paper, the number of hidden layers of BP-ANN is 3, and the number of hidden layer nodes is 4.

The training process of BP-ANN is:

(1) The weights and bias vectors of the neural network are initialized, and the weights and biases from the input layer to the hidden layer and the hidden layer to the output layer are denoted as $\omega^{(0)}$, $b_1^{(0)}$, $v^{(0)}$ and $b_2^{(0)}$, respectively.

(2) The forward propagation process of information is carried out to calculate the output value of each layer and its corresponding loss function:

$$E(\theta) = \frac{1}{n}\sum_{i=1}^{n}(y_i - \phi(v_i\phi(w_i + b_{i-1}) + b_i),\tag{15}$$

where $\theta$ represents the parameter collection of the neural network, $y_i$ represents the true value in the data, $\omega_i$ represents the weight of the neural network, $b_i$ represents the bias of the neural network and $\varphi$ represents the activation function;

(3) Calculate the error terms of the output and hidden elements based on the loss function. The error terms of the output unit are:

$$\nabla_{(k)}v = \frac{\partial E}{\partial v} = \frac{\partial z_2}{\partial v}\frac{\partial E}{\partial z_2}\frac{\partial E}{\partial p}\tag{16}$$

$$\nabla_{(k)}b_2 = \frac{\partial E}{\partial b_2} = \frac{\partial z_2}{\partial b_2}\frac{\partial p}{\partial z_2}\frac{\partial E}{\partial p},\tag{17}$$

The error terms of the hidden cell are:

$$\nabla_{(k)}w = \frac{\partial E}{\partial w} = \frac{\partial z_2}{\partial w}\frac{\partial h}{\partial z_1}\frac{\partial z_2}{\partial h}\frac{\partial p}{\partial z_2}\frac{\partial E}{\partial p}\tag{18}$$

$$\nabla_{(k)}b_1 = \frac{\partial E}{\partial b_1} = \frac{\partial z_1}{\partial b_1}\frac{\partial h}{\partial z_1}\frac{\partial z_2}{\partial h}\frac{\partial p}{\partial z_2}\frac{\partial E}{\partial p} \nabla_{(k)}b_1 = \frac{\partial E}{\partial b_1} = \frac{\partial z_1}{\partial b_1}\frac{\partial h}{\partial z_1}\frac{\partial z_2}{\partial h}\frac{\partial p}{\partial z_2}\frac{\partial E}{\partial p}, \tag{19}$$

where $z_1$ represents the input value of the hidden layer, $z_1$ represents the output layer, $h$ represents the output value of the hidden layer, and $p$ represents the model predicted value;

(4) Update weights and biases in the neural network. The updated output unit is:

$$v^{(k)} = v^{(k-1)} - \eta\nabla_{(k)}v \tag{20}$$

$$b_2^{(k)} = b_2^{(k-1)} - \eta\frac{\partial E}{\partial b_2}, \tag{21}$$

The updated hidden unit is:

$$w^{(k)} = w^{(k-1)} - \eta\nabla_{(k)}w \tag{22}$$

$$b_1^{(k)} = b_1^{(k-1)} - \eta\frac{\partial E}{\partial b_1}, \tag{23}$$

where $\eta$ represents the learning rate and $k$ represents the number of iterations;

(5) Repeat the above steps repeatedly, when the loss function is less than a given threshold or the number of iterations is greater than the set number of times, stop the iteration; this article believes that the parameters obtained at this time are the best parameters.

Before feeding data into the neural network, it is important to normalize the data. This can not only speed up the calculation of the neural network, but also improves the accuracy of the algorithm. In order to facilitate the calculation, this paper normalized the data to the interval of $[-1,1]$, and the normalization method is:

$$y = -1 + \frac{2(x - x_{\min})}{x_{max} - x_{\min}}, \tag{24}$$

where $x$ is the data before normalization and $y$ is the normalized data.

### 2.3. RBF Artificial Neural Network

Compared with BP artificial neural network, RBF artificial neural network is an effective feed forward neural network, which has significant advantages such as strong global approximation ability, no local minimum problems and fast learning speed [30]. It usually consists of an input layer, hidden layer, and output layer, in which it can be adjusted according to the actual need for the number of neurons in the hidden layer. In this paper, the number of RBF-ANN neurons in the hidden layer was the same as the number of input samples, and its network structure diagram is shown in Figure 3.

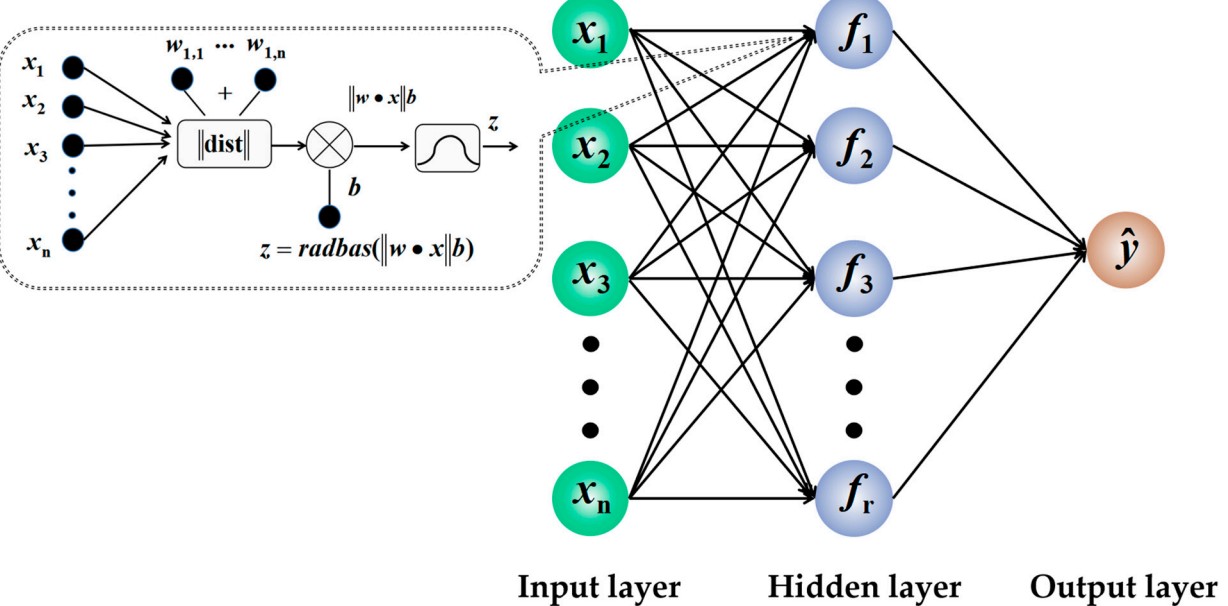

**Figure 3.** Structure diagram of RBF-ANN.

BP-ANN can have one or multiple hidden layers, while BRF-ANN have only one hidden layer. The first layer of the BRF-ANN is the input layer, which only plays the role of transmitting information and does not do any transformation processing on the input data. The second layer is a hidden layer, the number of nodes of the hidden layer is not fixed, it can be adjusted according to actual need, and based on the task goal is constantly changing, the activation function of the hidden layer is a non-negative linear function symmetrical along the center point and constantly decaying rapidly to both sides, with local response characteristics. The third layer is the output layer, which will linearly transform the input data and then the output.

The activation function of the hidden layer of the BP-ANN calculates the inner product of the input data and connection weights, while the independent variable of the activation function in the hidden layer of the BRF-ANN is the Euclidean distance between the input data and the center vector, and the activation function is the radial basis function. The farther the input data is from the center of the radial basis function, the less active it is. It can be seen that the output of the BRF-ANN is not related to all parameters, but only to a small number of parameters, and this article calls this characteristic of the BRF-ANN a local response characteristic. Therefore, BRF-ANNs are local approximation networks, while BP-ANNs are global approximation networks.

The hidden and output layers of a BP-ANN can be linear or nonlinear, while the hidden layer of an BRF-ANN is nonlinear, and the output layer is linear. The basic idea of BRF-ANN is that the radial basis function is used to construct a hidden layer space for the data in hidden nodes in the hidden layer, and the hidden layer converts the input data to a certain extent, and converts the low-dimensional mode input data into the high-dimensional space, so that the linear indivisible problem in the low-dimensional space becomes linearly separable in the high-dimensional space.

In the learning process of BRF-ANN, the most critical problem is how to determine the expansion coefficient of the hidden layer activation function. The common method is to select directly from a given set of training samples according to a certain method, or to determine by clustering. In this paper, the center point was randomly selected from the input sample using the direct selection method.

The activation function of the BRF-ANN is generally the Gaussian function:

$$f(x) = e^{-\frac{r^2}{2\sigma^2}}, \tag{25}$$

In the above equation, $r$ is the Euclidean distance of the input data to the center point, $\sigma$ represents the rate at which the function falls to 0, also known as the expansion factor. As can be seen from Figure 4, the smaller the expansion factor, the narrower the image.

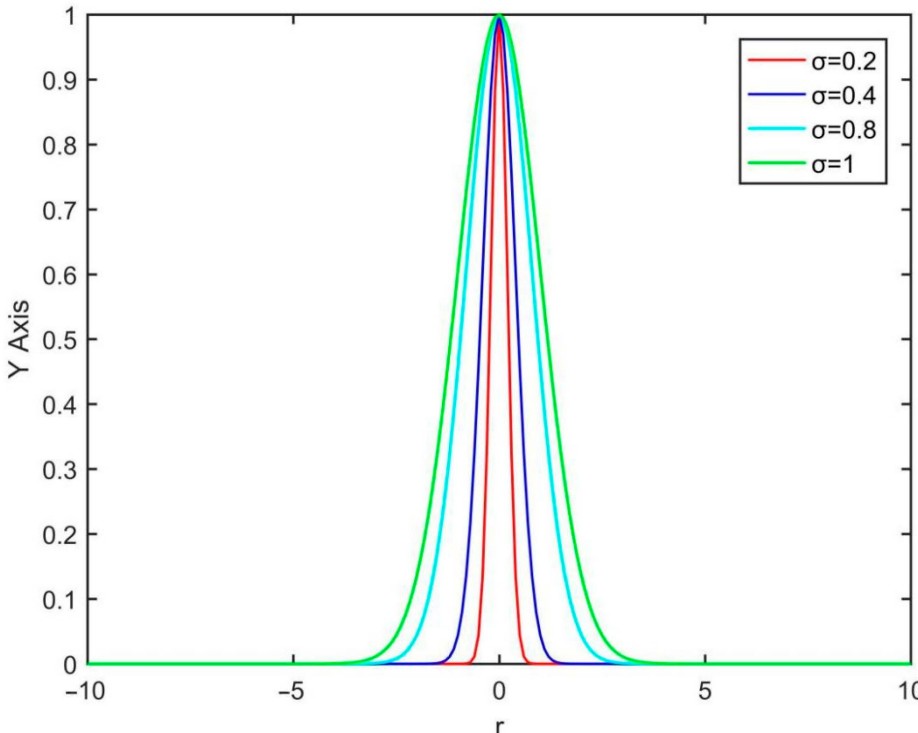

**Figure 4.** Gaussian radial basis function.

The training process of the BRF-ANN is as follows:

(1)   Determine the parameters. Initialize the connection weights between the hidden layer and the output layer:

$$w_{kj} = k_{\min} + j\frac{k_{\max} - k_{\min}}{q + 1}, \tag{26}$$

where $k_{min}$ is the minimum value of the output of the $k$-th neuron, $k_{max}$ is the maximum value of the output of the $k$-th neuron, and $q$ is the number of output layer units.

Initialize the center parameters of each neuron in the hidden layer: In order to reflect the characteristics of the input information to the greatest extent, the values of the centers of neurons in different hidden layers should be as different as possible and should correspond to the width vector. In order to show the characteristics of the input information more obviously, this paper changes the initial value of the central component of each neuron in the hidden layer from small to large equal spacing, so that the weaker input information produces a stronger response near the smaller center. The size of the spacing is determined by the number of neurons in the hidden layer. Finally, the initial value of the central parameter of each neuron in the hidden layer of the BRF-ANN can be expressed as:

$$c_{ji} = i_{\min} + \frac{i_{max} - i_{\min}}{2p} + (j - 1)\frac{i_{max} - i_{\min}}{p}, \tag{27}$$

where $i_{min}$ is the minimum value entered for the $i$-th feature value $i_{max}$ is the maximum value entered in the $i$-th feature, and $p$ is the number of neurons in the hidden layer.

Initialize the width vector: As can be seen from Figure 4, the smaller the width vector, the narrower the image of the activation function, and the smaller the response of other neuron centers in this neuron. Its calculation formula is:

$$d_{ji} = d_f \sqrt{\frac{1}{N} \sum_{k=1}^{N} (x_i^k - c_{ji})}, \tag{28}$$

(1) Input layer to hidden layer calculation:

$$z_i = e^{-\left\| \frac{x - c_j}{D_j} \right\|^2}, \tag{29}$$

where $c_j$ is the center vector corresponding to the *j*-th hidden layer neuron and $D_j$ is the width vector of the *j*-th hidden layer neuron.

(2) Calculation of the output layer:

$$y_k = \sum_{j=1}^{p} w_{kj} z_j, \tag{30}$$

(3) Updated iteration of weight:

$$w_{kj}(t) = w_{kj}(t-1) - \eta \frac{\partial E}{\partial w_{kj}(t-1)} + \alpha[w_{kj}(t-1) - w_{kj}(t-2)] \tag{31}$$

$$c_{ji}(t) = c_{ji}(t-1) - \eta \frac{\partial E}{\partial c_{ji}(t-1)} + \alpha[c_{ji}(t-1) - c_{ji}(t-2)] \tag{32}$$

$$d_{ji}(t) = d_{ji}(t-1) - \eta \frac{\partial E}{\partial d_{ji}(t-1)} + \alpha[d_{ji}(t-1) - d_{ji}(t-2)], \tag{33}$$

where is $\eta$ the learning rate and *E* represents the loss function.

In the training process, this paper first initializes the weight from the hidden layer to the output layer, the central parameters of each neuron in the hidden layer, and the width vector, and then calculates the loss function, when the loss function is less than the given threshold or the number of iterations is greater than the set number of times, stop the iteration, otherwise the gradient descent method is used to recalculate each weight until the conditions are met.

## 3. Data Simulation and Testing

### 3.1. Data Simulation

In order to solve the T-L model, this paper simulated the magnetic interference generated by the UAV during flight according to a standardized flight method (FOM) designed by Leliak and used the corresponding compensation algorithm to solve the simulated data, calculate the corresponding 16 compensation coefficients, and apply them to another set of data simulated by the standardized flight method to test the generalization ability of the model.

The FOM flight method is as follows: the aircraft flies sequentially in the order of north, east, south and west, each of which includes three $\pm 5°$ yaws, three $\pm 5°$ pitches and three $\pm 10°$ rolls, and the duration of each group of maneuvers is about 5~10 s, and 5 s of flat flight are interspersed between each group of maneuvers. The magnetic field in the flight area changes steadily, and in order to reduce the interference of shallow geological bodies on the aircraft, the flight altitude is generally set to 2000~3000 m.

This paper simulated two different sets of FOM flight data, denoted as Flight A and Flight B. Assuming that the aircraft is compensated in strict accordance with the FOM flight method, the geomagnetic field *T* is 46,862 nT, the magnetization inclination angle

is $-1.298°$, and the magnetization declination angle is $36.663°$, $T_{gx}$, $T_{gy}$ and $T_{gz}$ can be calculated according to Formulas (8)~(10), and then the fluxgate three-component, $T_{bx}$, $T_{by}$ and $T_{bz}$ can be calculated by correcting the obtained data according to Formulas (6) and (7). The three-axis fluxgate and the corresponding aeromagnetic interference model obtained by simulation are shown in Figures 5 and 6.

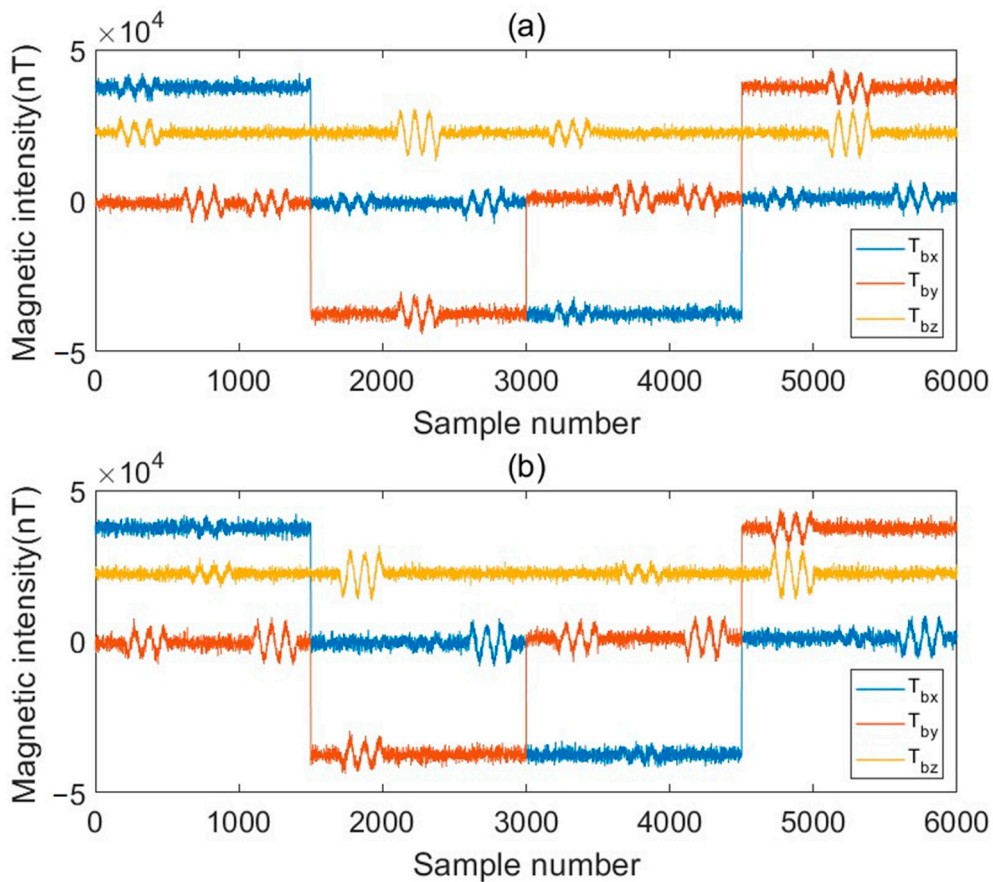

**Figure 5.** Simulation model of a three-axis fluxgate: (**a**) flight A; and (**b**) flight B.

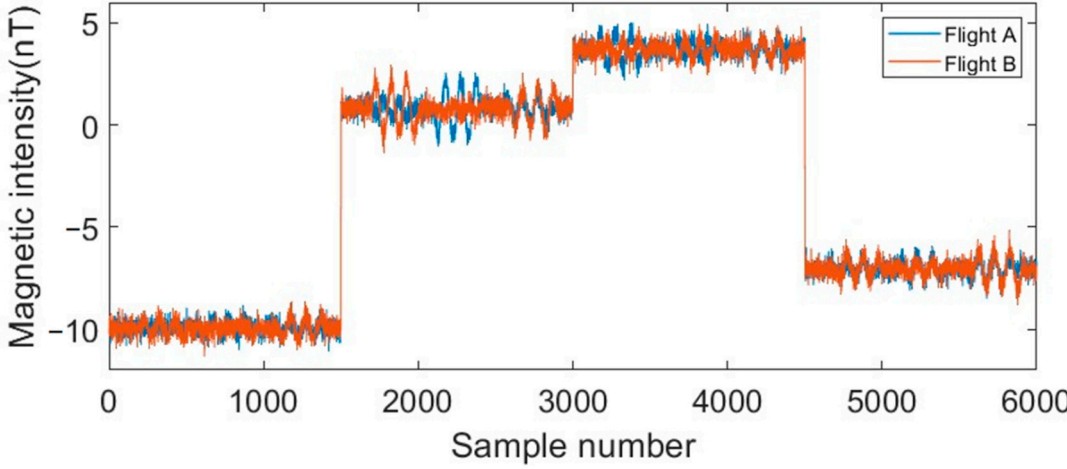

**Figure 6.** Aeromagnetic interference model corresponding to FOM flight.

### 3.2. Compensation Results

In order to verify the performance of the algorithm, this paper records two sets of simulation data, Flight A and Flight B, as training and test sets, respectively. As shown in Table 1, in the test set Flight A, the standard deviation of the raw data is 5.5873, the BP-ANN reduces the standard deviation to 0.0792, the IR is 70.5465, and the BRF-ANN reduces the standard deviation to 0.0649 and the IR is 86.1052. In the test set Flight B, the standard deviation of the raw data is 5.5927, the BP-ANN reduces the standard deviation to 0.0828 and the IR to 69.5160, and the BRF-ANN reduces the standard deviation to 0.0651 and the IR to 85.9473. The result after compensation is shown in Figure 7.

**Table 1.** Comparison of compensation results of BP-ANN and BRF-ANN.

| Test Set | Training Set | Method/Model | $STD_p$ | $STD_f$ | IR |
|----------|--------------|--------------|---------|---------|-----|
| Flight A | Flight B | BP-ANN | 5.5873 | 0.0792 | 70.5465 |
|          |          | BRF-ANN |        | 0.0649 | 86.1052 |
| Flight B | Flight A | BP-ANN | 5.5927 | 0.0828 | 67.5160 |
|          |          | RBF-ANN |        | 0.0651 | 85.9473 |

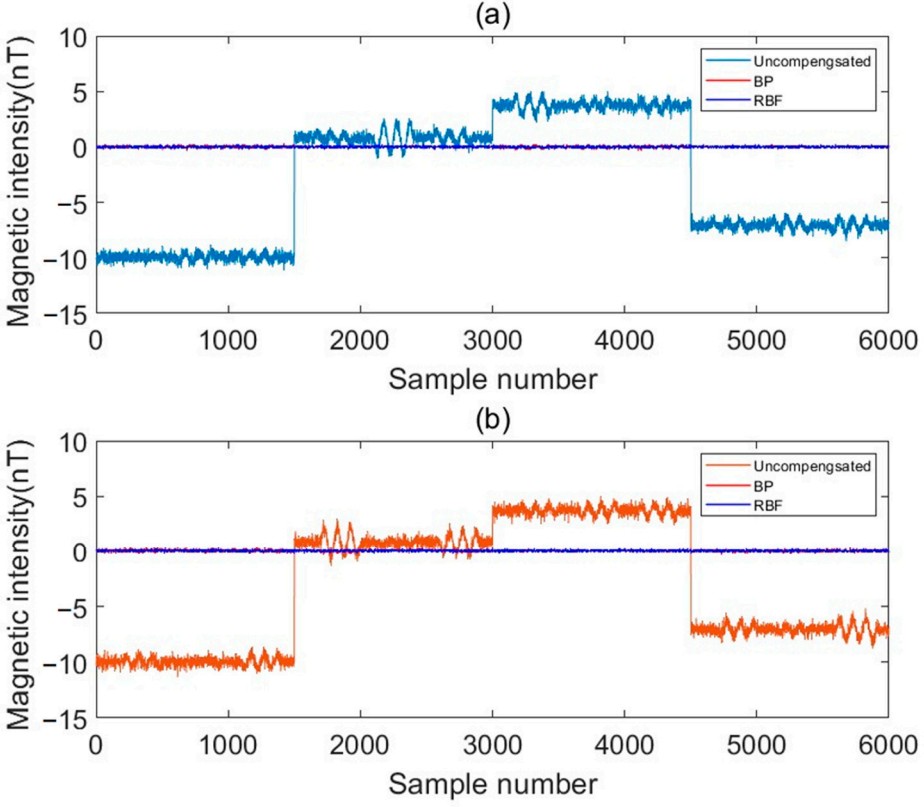

**Figure 7.** Simulation data compensation results: (**a**) test set flight A; and (**b**) test set flight B.

## 4. Real Measured Data Test

### 4.1. UAV Compensation Flight

In order to establish a suitable UAV aeromagnetic compensation model, this paper used a rotor UAV to carry out a compensation flight in Ma'anshan City, Anhui Province, China. The terrain is flat, there are no interference factors such as high-voltage power poles around, and the surrounding geomagnetic field does not change much, which is very in line with the hypothetical conditions set by the T–L model. This paper developed a UAV magnetic survey system, as shown in Figure 8, which is mainly composed of two parts: UAV platform and aeromagnetic survey equipment. The selected UAV platform is a

fuel-powered unmanned helicopter SU-H2M, which can take off and fly autonomously, and has the characteristics of long endurance, fast speeds, and a large cruising distance. The maximum payload is 45 kg, the battery life is 2 h, and the general cruising speed is 60 km/h. The aeromagnetic measurement equipment is mainly composed of five parts, as shown in Figure 9: (1) a high-precision potassium pump magnetometer (GSMP-35U used to measure the total magnetic field strength of the geomagnetic field; (2) a triaxial fluxgate magnetometer (TFM100-G2) used to record aircraft attitude change information; (3) a laser altimeter from the MDL company with a range of 0.05 m~200 m; (4) an inertial navigation module, using OEM 62 GPS locator, with a static plane positioning accuracy within ±2 m; and (5) the data collector and data processing platform, as shown in Figure 10 (detailed parameters are shown in Table 2).

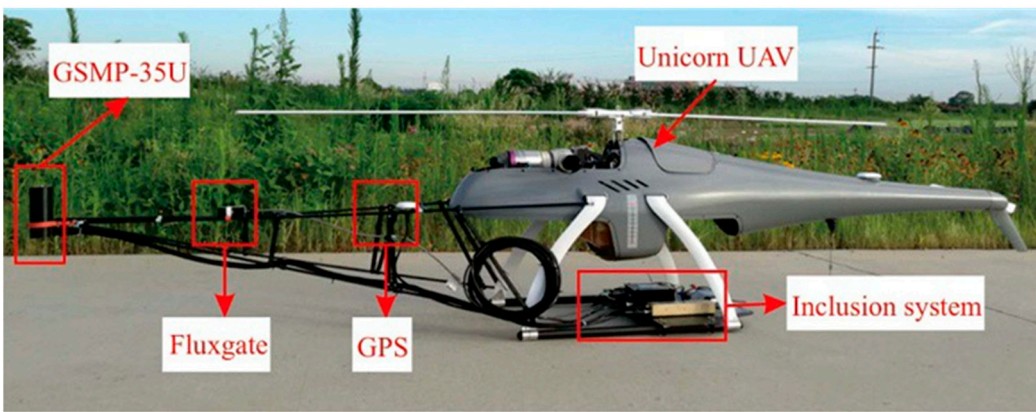

**Figure 8.** UAV detection platform.

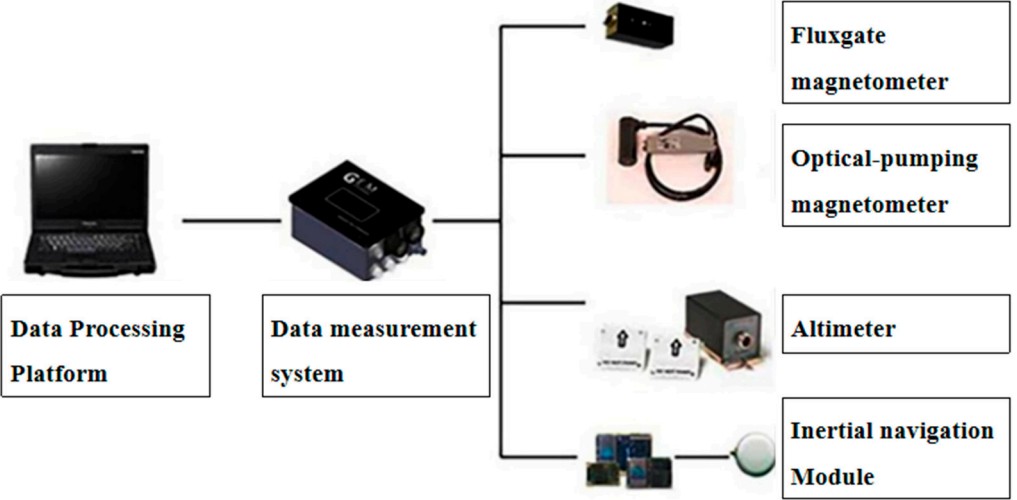

**Figure 9.** Aeromagnetic measurement equipment.

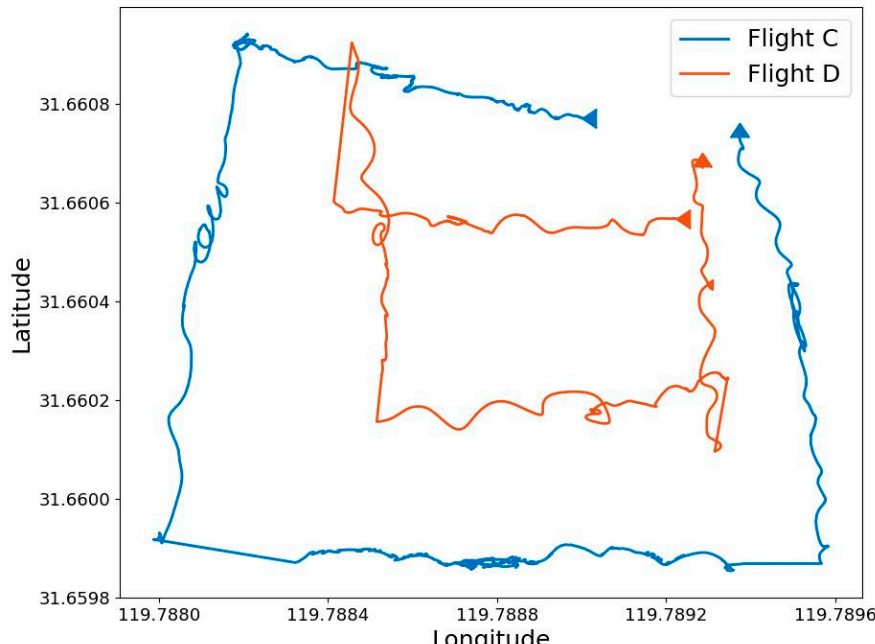

**Figure 10.** Flight C and Flight D flight paths.

**Table 2.** Main technical parameters of GSMP-35U high-precision potassium optical pump magnetometer.

| Measuring Range | 20,000 nT~120,000 nT |
| --- | --- |
| Gradient capacity | 50,000 nT/m |
| Sensitivity | 0.0003 nT@1Hz |
| Resolution | 0.0001 nT |
| Measurement accuracy | ±0.05 nT |
| Sample rate | 1, 2, 5, 10, 20 Hz |
| Operating temperature | −20 °C~+55 °C |

A total of two flights were carried out in this experiment, and the aeromagnetic interference data measured in the two compensation flight experiments are named Flight C and Flight D. The flight data of the two compensated flights are shown in Figures 10 and 11.

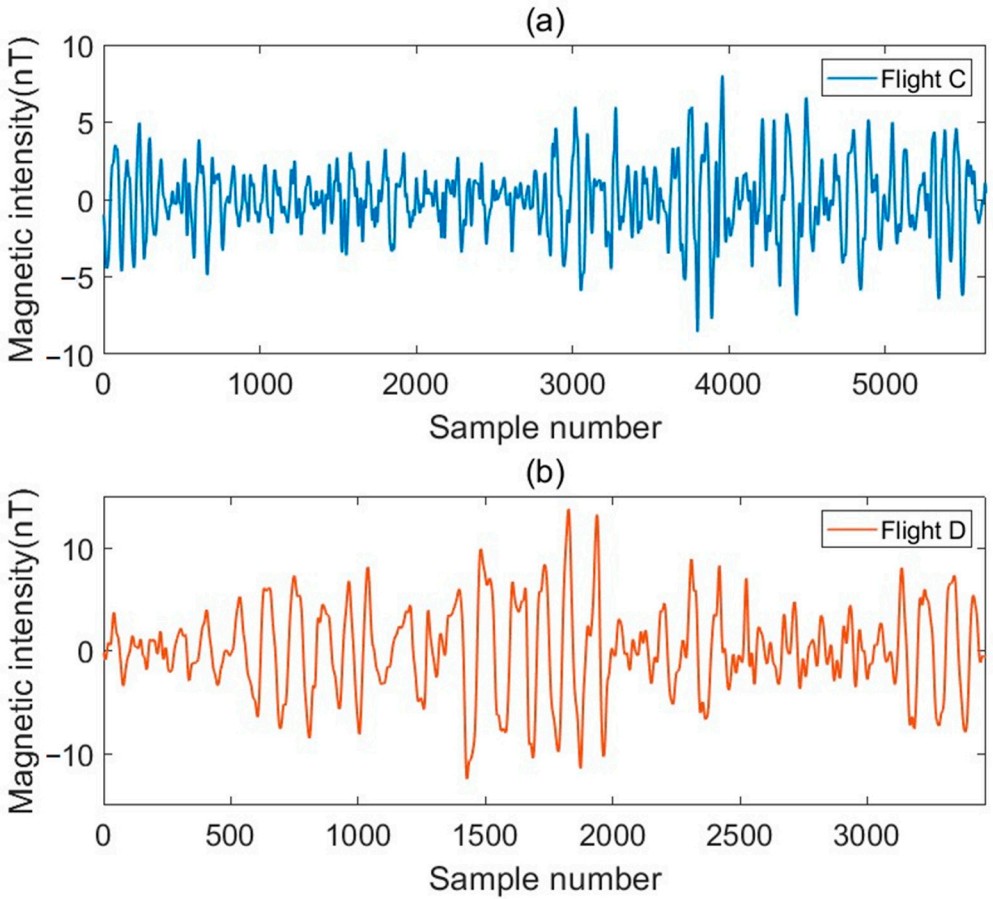

**Figure 11.** Aeromagnetic interference model: (**a**) Flight C; (**b**) Flight D.

*4.2. Compensation Results*

At present, the ratio (*IR*) of the standard deviation of magnetic interference before and after aeromagnetic compensation is commonly used in the industry to evaluate the compensation effect of aeromagnetic compensation methods.

$$IR = \frac{STD_p}{STD_f} \tag{34}$$

$$STD = \sqrt{\frac{1}{n}\sum_{i=1}^{n}(x_i - \mu)^2}, \tag{35}$$

where $STD_p$ is the standard deviation of the magnetic interference before compensation, and $STD_f$ is the standard deviation of the residual magnetic interference after compensation, $\mu$ is the arithmetic mean of the data.

In order to verify the aeromagnetic compensation effect of the above two methods, the data of Flight D and Flight C are used as training sets to compensate for Flight C and Flight D, respectively. The compensation result is shown in Figures 12 and 13. Table 3 shows the comparison of the compensation effects of BP-ANNs and BRF-ANNs. In Flight C, the BP-ANN reduced the standard deviation from 2.2804 to 0.3376 with an IR of 6.7547, and the BRF-ANN reduced the standard deviation from 2.2804 to 0.3091 with an IR of 7.3775. In Flight D, the BP-ANN reduced the standard deviation from 4.2558 to 0.5734 with an IR of 7.4220, and the BRF-ANN reduced the standard deviation from 4.25584 to 0.4734 with an IR of 8.9899. From the compensation results, it can be seen that the compensation effect of BRF-ANN is better than that of BP-ANN, which proves the superiority of BRF-ANN compensation method.

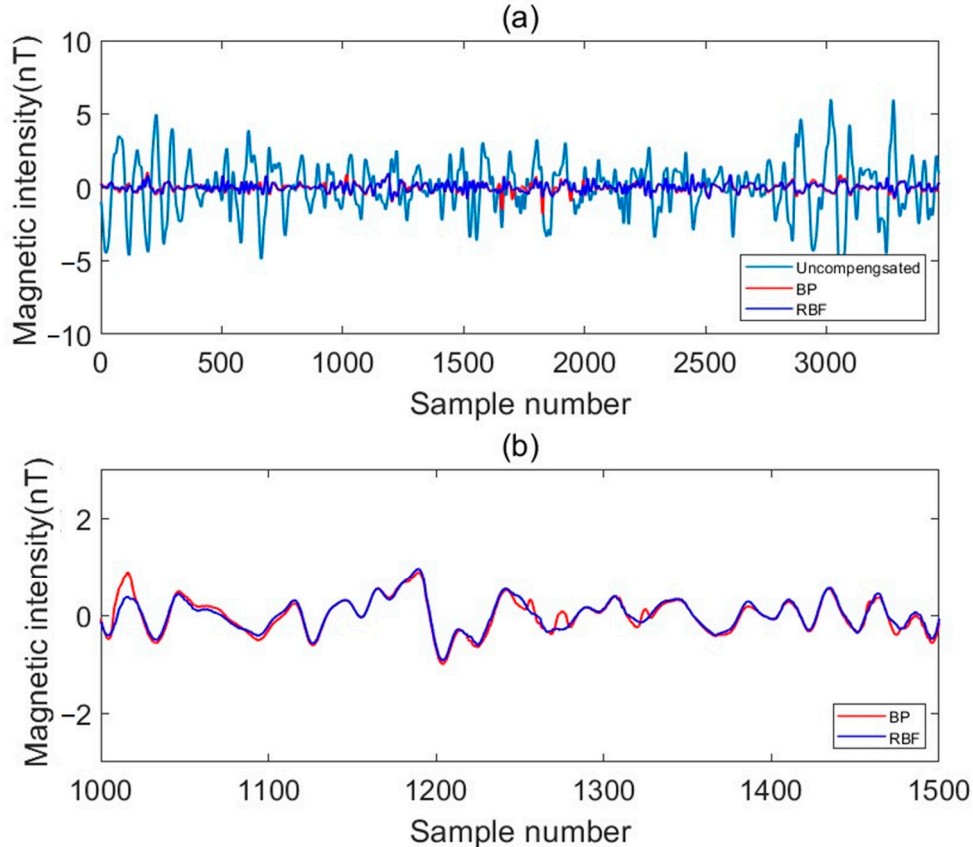

**Figure 12.** Flight C compensation results: (**a**) BP-ANN and BRF-ANN compensation results; and (**b**) compensation for the resulting sampling points 1000 to 1500.

**Table 3.** Comparison of compensation results of BP-ANN and BRF-ANN.

| Test Set | Training Set | Method/Model | $STD_p$ | $STD_f$ | IR |
|----------|--------------|--------------|---------|---------|-----|
| Flight C | Flight D | BP-ANN | 2.2804 | 0.3376 | 6.7547 |
|          |          | BRF-ANN |        | 0.3091 | 7.3775 |
| Flight D | Flight C | BP-ANN | 4.2558 | 0.5734 | 7.4220 |
|          |          | RBF-ANN |        | 0.4734 | 8.9899 |

This section may be divided by subheadings. It should provide a concise and precise description of the experimental results, their interpretation, as well as the experimental conclusions that can be drawn.

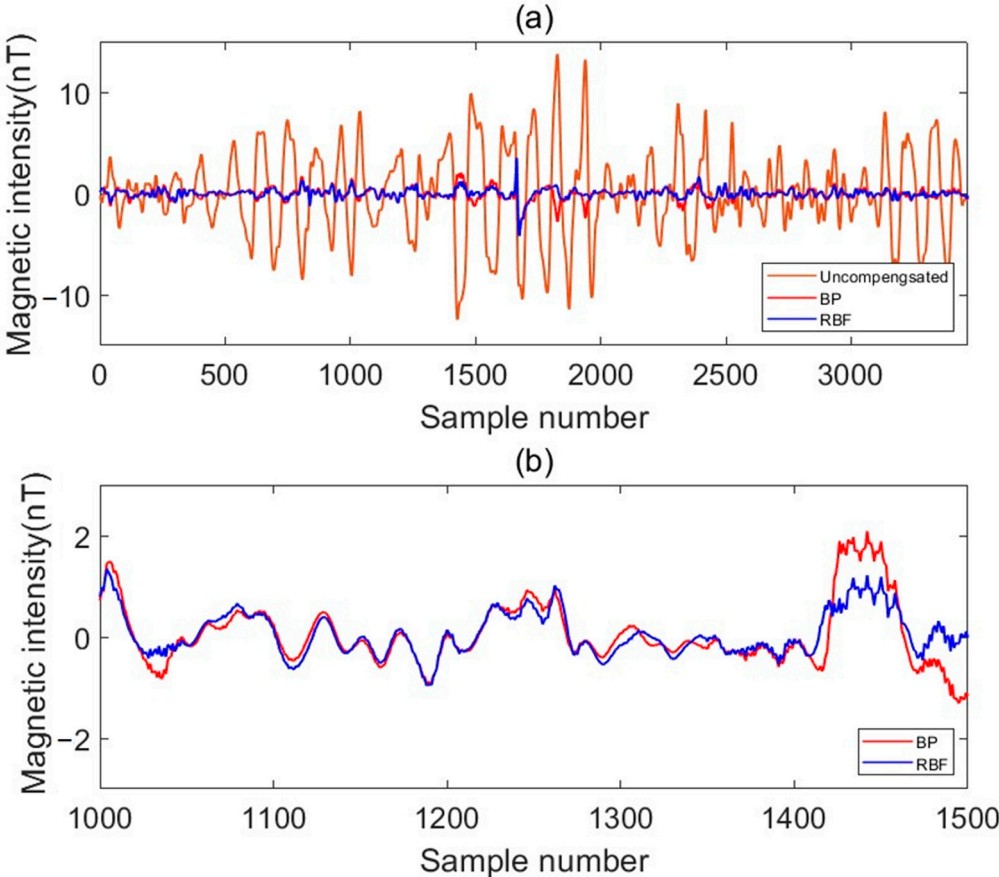

**Figure 13.** Flight D compensation results: (**a**) BP-ANN and BRF-ANN compensation results; and (**b**) compensation for the resulting sampling points 1000 to 1500.

## 5. Conclusions

In the previous aeromagnetic compensation work, although the classical BP-ANN is stronger than traditional regression algorithms in terms of fitting ability, the BP-ANN is a global approximation network, with limited generalization ability, and there are problems, such as falling into a local minimum easily, gradient disappearance, and an overfitting problem in magnetic compensation which affects the accuracy of aeromagnetic compensation. In order to improve the accuracy of compensation, on the basis of the T–L model, we proposed a compensation algorithm based on BRF-ANN, in which the hidden layer node uses the distance between the input mode and the central vector as the independent variable of the function, and uses the radial basis function as the activation function, which has the characteristics of local approximation and better generalization ability, avoids the problem of local minimum effectively, and improves the accuracy of magnetic compensation to a certain extent. We verified the feasibility of this method in simulated data and measured data experiments.

**Author Contributions:** Z.S., J.J. and C.Y.: Conceptualization, Methodology, Software; J.J. and S.Z.: Resources Data curation; C.Y.: Writing—Original draft preparation; C.Y. and Z.S.: Visualization; Z.S., C.Y., P.Y. and S.Z.: Writing—Reviewing and Editing. All authors have read and agreed to the published version of the manuscript.

**Funding:** This work was supported by National Key R&D Program of China (No. 2020YFE0201300), National Natural Science Foundation of China (No. 42204141), the Natural Science Foundation of Jilin Province (No. 20210508033RQ), National Natural Science Foundation of China (No. 42274187), Interdisciplinary training Program for Young teachers and students of Jilin University (415010300086).

**Conflicts of Interest:** The authors declare no conflict of interest.

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
