# Peer review of "An Aeromagnetic Compensation Algorithm Based on Radial Basis Function Artificial Neural Network"

_applsci, doi:10.3390/app13010136_

Round 1

Reviewer 1 Report

The reviewed article is scientific interest, since it is devoted to discussing the problems of processing data from aeromagnetic surveys from UAV and improving the accuracy of measuring aeromagnetic data using the proposed radial basis function (RBF) neural network algorithm to compensate for them. The title of the article and keywords adequately reflect its content. In the abstract, the authors give the essence of the problem and its state, describe the methods and models of the study, and the results obtained.

In the introduction, the authors provide a brief literature review on the article topic, formulate the study goal and the features of the proposed method. The second section is devoted to describing the models and methods used to solve the described problem. The third part is devoted to the description of data modelling and algorithm testing processes. In the fourth section, the authors present the results of testing real measured data. In the "Conclusions" section, the authors summarize the results obtained and draw conclusions on the work.

The article is prepared in accordance with the instructions for the authors, corresponds to the topic that it explores and publishes. In our opinion, the article is in line with the theme of “Improving the quality of mineral exploration with the help of UAVs” and corresponds in type to the Preliminary Study.

Comment.

1.      The topic is interesting, but the literature review should be more thorough, so it would be possible to indicate what are the shortcomings of the methods proposed in the reviewed articles and why these methods are not rational for solving the problem identified by the authors.

2.      At the end of the first chapter, the authors indicate the difference between the proposed method and the existing ones and its novelty, but do not give the conditions for application, limitations and practical significance. It would also be nice if the authors indicated in first section the article structure.

3.      In general, it is not clear how an article that has a relevant and significant topic reflects its scientific and practical component, in particular, who is interested in applying research results in practice and how this will improve the efficiency of processes.

4.      It is necessary to check the numbering of figures and tables (After the figure with number 13, there is again figure number 12, and after the table with number 2, again table number 1).

Reviewer 2 Report

Title and abstract of paper promises really fun in reading, but….

Section 2.2.1. and 2.2.2 -> The description of BP type ANN is redundant, it is enough to provide a literature reference. As athors postulate in abstract: “Compared with the classical backpropagation (BP) neural network, the test results of the sythnetic data and real measured magentic data showed that the RBF neural network has higher compensation accuracy and stronger generalization ability.”

Beside, in my opinion “RBF neural network” looks odd. I like more ANN type RBF.

Moreover authors write:

“BP neural networks can have one or multiple hidden layers, while RBF neural networks have only one hidden layer.” In view of the above, the RBFs are the special case of BP. Thus the whole article has to be corrected to obtain clarity of text and results presentation. Beside what is difference of BP with MLP? Then why BP is used instead of standard MLP name of ANN type? BP is more an algorithm than ANN topology.

Table 1 – I can see another type of ANN: “BBF”.

In addition, I would like to see the topology of the networks being compared: this is the number of inputs, the number of neurons in successive layers and their association with signals, and the method of sampling the signals. I would also like to see an electronic system in which the network is built-in.

This is the result of authors statement: “The three-axis fluxgate and the corresponding aeromagnetic interference model obtained by simulation are shown in Figure 6 and Figure 7.” Thus I can guess that proposed work is simulation over simulation type and the abstract and title promises presentation of real investigation results analysis. In my opinion such type of initial data have to result in “Smart random number generator used to prove postulated thesis” title of paper.

Please provide real data for such fascinating experiment. Such method is quite popular. Such method is quite popular. See MDPI journals publications as an example of real data use,  these positions can be referred :

M. Borecki, A. Rychlik, O. Vrublevskyi, A. Olejnik, M.L. Korwin-Pawlowski, Method of Non-Invasive determination of wheel rim technical condition using vibration measurement and artificial neural network, Measurement, Volume 185, 2021, 110050, ISSN 0263-2241, https://doi.org/10.1016/j.measurement.2021.110050.

Popela, M.; Leuchter, J.; Olivová, J.; Richterová, M. Development of a Remote-Controlled Electrical Interference Vehicle with a Magnetron. Sensors 2020, 20, 6309. https://doi.org/10.3390/s20216309

Azam, S.; Munir, F.; Sheri, A.M.; Kim, J.; Jeon, M. System, Design and Experimental Validation of Autonomous Vehicle in an Unconstrained Environment. Sensors 2020, 20, 5999. https://doi.org/10.3390/s20215999

And please read what you write, before next submission.

Reviewer 3 Report

This paper uses the RBF neural network for aeromagnetic compensation. The application of this paper is very useful for aerospace engineering. However, the weak point of this paper is to present the originality of this work because the RBF function is one of the famous neural network functions. The authors can increase the originality of this paper by showing some special problems of the application. These is my comments:

1) This paper has some typo errors such as in Table 1 the BBF must be RBF.

2) How many hidden layers of the BF neural network? It is better to show more detail about this information.

3) Is it possible to show how to select the number of hidden layers for the BF method? This is very important to show whether your BF is good enough to solve this problem or not.

4) It better to do more review about the application of the RBF neural network for the aerospace application such as the following paper:

[1]Phiboon, Tharathep, et al. "Experiment and computation multi-fidelity multi-objective airfoil design optimization of fixed-wing UAV." Journal of Mechanical Science and Technology 35.9 (2021): 4065-4072.

Round 2

Reviewer 2 Report

Accept in current form.

Author Response

Thank you so much. 

Reviewer 3 Report

The authors can answer all of my question. Now this paper is suitable to published.

Author Response

Thank you so much.